# Exploring High vs. Low Burnout amongst Public Sector Educators: COVID-19 Antecedents and Profiles

**DOI:** 10.3390/ijerph19020780

**Published:** 2022-01-11

**Authors:** Ana Pérez-Luño, Miriam Díez Piñol, Simon L. Dolan

**Affiliations:** 1Business Administration Department, Pablo de Olavide University, Carretera de Utrera Km 13, 41013 Seville, Spain; 2Global Future of Work Foundation, 08005 Barcelona, Spain; diezpinolm@gmail.com (M.D.P.); simon@globalfutureofwork.com (S.L.D.)

**Keywords:** work-related sources of stress, family-related sources of stress, gender, support and control, burnout, resilience, COVID-19 era, educators

## Abstract

The COVID-19 pandemic has had a prolonged impact on many people working in different sectors. This paper focuses on the psychological stress consequences of professionals working in the educational sector in Andalucía (Spain). Using a sample of 340 educators, this empirical paper identifies the antecedents and profiles of those that ended up with burnout vs. those that were able to develop resilience. Results from OLS regressions show that regardless of the origins of stress, the principal determinant of burnout is clearly a lack of support and a perception of an inability to control a situation. Furthermore, results also show that working sources have a higher impact on the configuration of high burnout, while family sources harm those who are more resilient (low burnout).

## 1. Introduction

We live in a very turbulent world, which is referred to by many as VUCA (Volatile, Uncertain, Complex-Chaotic and Ambiguous). In addition, 2020–2021 will be remembered as the years containing COVID-19 pandemic, which has contributed even more to this situation. Ample research and journalistic anecdotes show that the proportion of people in various sectors suffering from this situation has dramatically risen. This situation has been even more dramatical in the education sector [1]. Some research suggests that stress has become endemic [2]. An interest in studying stress is not a new phenomenon. The term has been used since the 14th century to refer to negative experiences, such as adversity, difficulties, suffering, affliction, etc. [2,3]. Therefore, the general aim of this study is to explore the principal determinants of stress on professionals in the education sector during the almost 2-year COVID-19 period.

In the literature, there have been some studies exploring issues related to the wellbeing and mental health of different types of service professions. One of the most studied professions has been teachers [4]. In modern societies, the professional function of teachers has been questioned [4,5,6], receiving ample environmental pressures. According to the European Foundation for the Improvement of Living and Working Conditions [7], prior to the pandemic teachers across Europe had been protesting about their working conditions (payment, working time and workload, among others). For those who have studied chronic stress (at work) for many years, the lockdown situation has presented an ideal laboratory not only to better understand the phenomenon of working people, but also to be inspired by the growing literature on positive psychology and identify those who suffered less and became resilient compared to those that ended up much worse as a result [7,8]. The empirical results of such a study may shed light on people, conditions, and circumstances where psychological stress can be better managed and where resilience can be enhanced [9]. 

The last survey on occupational risks in Spain [10], showed that companies find it more complex to deal with psychosocial risks than physical ones. It was also seen that the predominant sectors applying measures to prevent psychosocial risks are health, education, and social services. In the workplace, the concern of researchers was placed on identifying the antecedents and consequences of stress amongst professionals of different sectors [11,12]. According to [4], almost 22% of the studies about burnout in EEUU from 1978 to 1996 were focused on teachers. Understanding this complex and dynamic phenomenon allows us to understand people’s behaviors and undertake preventative measures that may help alleviate stressors and improve wellbeing on the one hand, and productivity on the other hand [13].

In industrialized societies, stress has all too often been interpreted as an unequivocal consequence of success (both personal and professional), although many scholars are warning us about these negative results and offer remedies to succeed without experiencing stressed [14]. The first scientific investigations on the psychological consequence of stress were found in the works of Freudenberg [15,16] and Maslach [17], who have defined and explained the notion of burnout. Longitudinal studies that have been carried out by Shirom [18] have shown that this type of issue remains stable over time, thus justifying the chronicity of its nature [19,20,21,22].

Traditional research in the past aimed at developing reliable models and measures that would permit the detection of chronic stress. The latter is problematic. Hence, the symptoms and signs are not so easy to detect and measure, despite the fact that its impact on the physical and psychological health of workers was well documented [23]. By contrast, it is easier to detect extreme cases of acute stress, hence some of the signs and symptoms are observable. The World Health Organization [24] provided data that showed that every 40 s, someone in the world has committed suicide due to depression and anxiety derived from either acute or long-term chronic stress. Therefore, in order to improve diagnosis and embark on pre-mental disease intervention, a reliable model and corresponding diagnostic tools are sought. In this sense, thousands of publications geared towards measuring and modelling chronic stress have been published, where both alterations in mental state such as depression or burnout were used. Hence, these may lead to suicide, as well as somatic ailments and a host of physical illnesses [25]. An exhaustive literature review shows the emergence of new “terms” in the field of stress such as mental or physical exhaustion, stress prone personalities (i.e., Type A behavior) and alike [26].

Of the different available models, a decision has been made in this paper to develop some creative and innovative angles to measure the state of mental health. In order to analyze chronic stress, this paper measures mental health in three complementary manners all connected to the concept of burnout: *its frequency, its severity, and its density*. The latter is an innovative algorithm that multiplies the frequency by its severity [27,28]. We have also analyzed different sources of stress and have identified them into two distinct categories: work and family. According to the European Agency for Safety and Health at Work [29], work-related sources of stress are the second highest work-related health problem. In Spain, for example, a high number of workers consider that work stress is frequent and that their organizations do not manage it properly [30]. Moreover, during the pandemic, some argued that stress levels have been elevated to 24% [31,32]. Thus, one of the interesting angles of research advanced herein is the examination: if work compared to family stressors play a greater role in its relative impact on burnout.

One needs to remember that COVID-19 related confinement has meant that teleworking has been imposed by many educational institutions prematurely as the employees were not sufficiently trained [1]. Many of them relayed professional frustration for the lack of resources provided by the corresponding employers, which would enable them to operate more efficiently in a virtual manner. In the case of the educational sector, teaching methods and study plans rapidly moved from the traditional physical environment to one that is virtual, and all of this took place at record speed. As a consequence, it is hypothesized that a substantial proportion of professionals in this educational sector have experienced high emotional impact, suffering silently and anonymously in many instances. The fact is that the vast majority of educational centers were closed, new methodologies such as an online modality were required, pressure from families and the students alike was felt in demanding solutions to special educational needs [33,34]. 

Based on the research above, the first aim of this study is to explore the principal determinants of burnout of professionals in the education sector during the almost 2-year COVID-19 period. The second objective of this paper is to pinpoint the relative impact of origins of stress connected to work settings and demands compared to family settings and demands. Numerous sources of occupational stress were identified by the literature and were incorporated in this study. We were inspired by the work of Cooper et al. [35,36]. In relative terms, there is ample research and publications published on the workplace setting and to a lesser extent, on the family environment (outside of work albeit the fact that the latter has gained attention during the pandemic) [37,38]. Nonetheless, this identification is important since the measures to be adopted by public officials will differ according to the origin of the stress. While the action and design of preventive measures at work are under the control of the educational institution, those related to the family environment are left to each individual. It became even more important to study stress at the family level, hence virtual work keeps many educational professionals working from home. A host of studies clearly suggest that about two-thirds of all employees experience a spillover effect (stress at work affects family, and stress at family affects work) [39,40,41].

The third objective of the study is to pinpoint the impact of moderators, key individual differences, and control variables, that can explain the direct effects of the sources/origins of stress on psychological burnout. In this regard, a comprehensive literature review was undertaken, and the relevant variables were identified and included in our proposed model (Figure 1). Among others, notes worth mentioning include: the perception of the support received, the perceived control over resources, and a host of social demographic characteristics such as age, gender, educational level, having children, practicing sport and other hobbies, and more. 

According to Cooper [42], employees need to be supported and rewarded, but they also want to be part of a purposeful place of work and to feel part of a community. We hope that completing these objectives will help the scholarly community and practitioners alike to help education professionals in managing stressful situations and identifying strengths and weaknesses in family and/or work settings. 

## 2. Methods

### 2.1. Sample

We surveyed personnel working in the education system in Andalusia. There were 107,269 educators at the time of data collection. From them, about 67% are women. We sent the questionnaire to all of the population. Out of these, 340 questionnaires were completed and represent the sample for which all analyses were affected. We tested the non-response bias and found no statistically significant differences for age, gender, and marital status. Moreover, our sample is representative of the population in terms of proportion of men and men, 64% are women in our sample and 65% in the full population. 

### 2.2. Measures

Many of the constructs included in the study were measured with multi-item scales as they were adapted from classical validated instruments developed over the years by Dolan and his colleagues (e.g., [43,44]). All of the measures are composed of multi-items and meet the rigor of all psychometric criteria. First, we pretested all of the measures in 10 interviews with people involved in the education system and asked them to closely review the survey, to ensure the clarity of the questions, and to ascertain whether or not the scales captured the desired information. We then revised any potentially confusing items before submitting the final questionnaire. 

#### 2.2.1. Dependent Variable: Psychological Burnout

The three dimensions of the classical MBI were employed here. MBI stands for the Maslach Burnout Inventory (MBI), which is the most popular psychological assessment instrument comprising 22 items. The original form of the MBI was developed by Maslach and Jackson [45] with the goal of assessing an individual’s experience of burnout. However, in this study, instead of using the 22 items of the standard MBI, we used a proxy measure of each of the dimensions of the MBI: Emotional and Physical Exhaustion, Depersonalization, and Lack of Personal Accomplishment or Motivation. Due to considerations connected with the overall length of the survey, only 4 items were used. The items were: (1) In the last 4 months I have felt physically tired (i.e., physically exhausted); (2) In the last 4 months I have felt low or no motivation to initiate any new project; (3) In the last 4 months I have felt emotionally exhausted; (4) In the last 4 months I have felt indifferent and/or apathetic to people around me. This short measure has been used for over 10 years with executives in stress management seminars, and content validity has been very satisfactory. A small study with MBA students where the original MBI and this proxy measure was used, yield a correlation of 0.83 and 0.93 with the EE dimension.

Another innovation in this study was to employ the burnout measure in 3 different manners: its frequency (the measure shows satisfactory reliability with the Cronbach’s alpha of 0.84), its severity (the measure shows satisfactory reliability with the Cronbach’s alpha of 0.839), and its density (the multiplication of both). The scales run from “Never (1); Once a month or less (2); A few times a month (3); Once a week (4); A few times a week (5); to Every day (6)” for frequency and from “Nothing (1); Very little (2); Little (3); Medium (4); Strong (5); to Very Strong (6)”.

Our proposal for identifying and separating the high burnout group from the low burnout group is worth mentioning. The sample was divided into two groups based on their mean score on the burnout scale. We used a split that is further explained in the results section (hereafter), but we wish to stress the importance of configurating the low burnout people (those who have low score on burnout), that we propose to label them as “resilient people”. Resilience is an emerging concept and for many, represent the other pole of stress. Resilience involves two defining elements—the experience of adversity and positive adaptation despite this adversity [46]. First, entities must experience adversity to demonstrate and build resilience. Adversity ranges on many continuums, including severity, duration, and predictability, which impacts individuals’ ability to demonstrate resilience. What matters, is how people perceive the adversity. Resilient people view adversity as a springboard for growth and development as opposed to a negative experience that should be avoided at all costs.

#### 2.2.2. Independent and Moderator Variables

We developed a *Composite index of Work Sources of Stress (hereafter C.W.S)* with a 7-item scale. Participant were asked to what extent they experienced the following in the past 4 months. (1) I felt that I had too many things to accomplish at work which measures work overload; (2) I felt that I was being underpaid for the type of work I do, which measures pay inequity; (3) I thought that my working conditions were not adequate which measures poor working conditions; (4) I felt that my superior or colleagues acted aggressively towards me, which measures aggressiveness at work; (5) I felt that my job/work was not made clear, and I didn’t understand what is expected from me which measures job ambiguity; (6) I felt that I had experienced conflicts at work, (either with my team, my colleagues, superior or clients) which measures job conflicts; (7) I sensed that I had too much responsibility in my work, which measures over-responsibility. The composite scale shows satisfactory reliability with the Cronbach’s alpha of 0.85.

We also developed a composite of *Family Sources of Stress (hereafter C.F.S)* with a 5-items. Participants were asked to what extent they experienced the following in the past 4 months. (1) I felt that my partner (husband/wife) had been aggressive with me, which measures aggressiveness at home; (2) I felt that I was being asked to be a super-person (superman-superwomen) at home and felt great overload of chores to do, which measure role overload at home; (3) I felt that I had been unappreciated by my partner, family members, relatives or friends, which measures lack of appreciation at home; (4) I felt helpless with my partner as there were constant conflicts between my work responsibilities and family responsibilities, which measures constant work-family conflicts; and (5) I felt overwhelmed by all sort of problems we were having at home, (partner, children, parents) which measures a feeling of being overwhelmed at home. The composite index shows satisfactory reliability with the Cronbach’s alpha of 0.81.

We have created a composite index measuring lack of support and lack of control (both work and home, hereafter and in the regression, results abbreviated to “C. Non-S&C”) with a 10-item scale. Participants were asked to what extent they experienced the following in the past 4 months. (1) I did not control and could not meet the expectations of my family; (2) I did not have enough resources to offer my family good quality of life, compared to my neighbors or friends; (3) I did not have any control over my work (i.e., feeling overwhelmed at work); (4) I truly felt that my destiny and success depended more on fate and luck rather than my own doings; (5) My partner (spouse) did not provide the kind of emotional and physical support that I really needed; (6) I did not get enough support from my superiors at work; (7) I did not get enough support from my colleagues at work (8) I did not get the support needed from my friends; (9) I did not get the support needed from my neighbors or my community; and (10) I did not get enough support from my family. The composite index shows satisfactory reliability with the Cronbach’s alpha of 0.89.

All of the scales for the independent and moderator variables run from “Never (1); Once a month or less (2); A few times a month (3); A few times a week (4); to Every day (5)”.

#### 2.2.3. Control Variables

We included several control variables in the analysis to eliminate possible cofounding effects on burnout. These included:

*Age*, has been introduced as a continuous variable that has been calculated by subtracting the interviewee’s date of birth from 2021 (year of the study). *Children* was introduced as a dummy variable with 1 representing having children and 0 representing not having children. *Other dependent*, another dummy variable with 1 representing having other people (not children) to take care of and 0 representing not having other relatives to take care of. We asked, “Has the way you work changed in the last 4 months?”. This variable has been labelled *Working change* and it is another dummy variable with 1 representing changes in the way work is developed and 0 representing the opposite. We also asked the professional if they practiced sports during the pandemic. This variable is labelled *Sports activities*, and it is a dummy variable with 1 representing that the individual practices sports and 0 representing the opposite. We also controlled for the type of education in which the professionals were involved (where they performed their professional activity), giving rise to the following variables: *secondary school, primary school, years 3 to 5, professional education and other education years*. We also controlled for having an extra position in a variable called *Controller*. This is a dummy variable with 1 representing having an extra position and 0 representing not having an extra position. Finally, we controlled for *Gender* and *Type A personality*. *Gender* is a dummy variable with 1 representing women and 0 representing men. We measured *Type A personality* with a 5 items scale. We asked participants, over the past 4 months, to what extent they (1) are an extremely competitive person; (2) really do not enjoy people stepping into their territory; (3) do everything fast, because they hate to waste time or wait; (4) really enjoy simultaneously tackle many tasks and responsibilities; (5) get really upset when people who work with them do not complete a task to perfection. The measure shows relatively satisfactory reliability with the Cronbach’s alpha of 0.64.

## 3. Results

Table 1 shows the descriptive statistics for all of the variables in our study including their means, standard deviations, minimum, maximum and correlations. Value Inflation Factors (VIFs) were computed (albeit not reported due to space limitations). None of them reached the score of above 4, indicating that we did not encounter any problems of multicollinearity. As Table 1 shows, burnout is correlated to all of the independent and moderator variables.

Data was primarily analyzed using an ordinary least square (OLS) procedure. In addition, in order to examine possible differences in patterns amongst high and low burned-out people, we use seemingly unrelated regression (SUR) with separate regression equations for each subsample (see Figure 2, Figure 3 and Figure 4 and Table 2, Table 3 and Table 4). The threshold to distinguish between low (i.e., resilient people hereafter) and high burnout, is based on the mean value. The results obtained with the mean values are similar to those obtained with the median values (not presented here due to space limitations). Specifically, we ran three sets of regression equations for the subsamples. The three sets contain two regression equations. In the first case, one regression was based on the observations of those who showed a low frequency of burnout (164 individuals), and the other was based on the observations of those who showed a high frequency of burnout (176 individuals). The second set contains observations of those who showed low severity of burnout (164 individuals), and the other was based on the observations of those who showed high severity of burnout (176 individuals). Finally, the third set contains observations of those who showed a low density of burnout (164 individuals), and the other was based on the observations of those who showed a high density of burnout (176 individuals).

Our results are quite consistent in their findings. That is, for the “low burnout-resilient” sample, whether we use frequency, severity, or density, we find that women have suffered more burnout than men, and that the absence of support and control and the composite of family sources have significantly contributed to increased burnout. We also find an interactive effect between the composite of work sources of stress and the absence of support and control. Figure 2 shows this interaction effect for low-frequency, low-severity and low-density of burnout. As it can be observed, in all cases, the perception of not having support or feeling in control make people more burned out. However, this stressful situation decreases as people have more work sources of stress. However, those with a low perception of absence of support and control are less burned out with low work sources, but their stress increases as work sources increase. A possible explanation for the finding that high work source reduces burnout for people with a high perception of non-support and control could be that these people find more meaning in what they do as work sources increase.

Again, our results are very consistent for the “high burnout” sample. Whether we use frequency, severity, or density, we find that people are more burned-out as they get older and that having children to take care of reduces the severity and density of burnout. As in the low burnout sample, the absence of support and control increases burnout. Finally, we find a moderating effect between family sources and absence of support and control. Figure 3 shows this interaction effects low-frequency, low-severity, and low-density of burnout. As it can be observed, in all cases and similar to what we found in the previous interaction, the perception of not having support or feeling in control makes people more burned out, and this increases as family sources increase. However, those with a low perception of absence of support and control, are less burned out as family sources increase. A possible explanation for this finding could be that people with the low perception of non-support and control, feel even better as their family sources increase. Figure 4 summarizes all of these findings.

## 4. Discussion

Given the complexity of the study design (low and high burnout group) and the multiple manners for which psychological burnout was measured, the discussion is rather complex. Hence, the configurations are cumbersome, and in the end, everything is relative. Thus, in order to avoid redundancy, and in the aim of simplicity, an attempt is made to focus on some implications and possible solutions that might benefit both the educational system establishment as well as the educators themselves.

By and large, the study shows the strength of using a new (more precise) algorithm to measure psychological burnout. While the traditional way to do that focused on frequency alone [47,48,49], and others focused on severity of each dimension alone [50,51,52,53,54], it seems logical to use a combined index of density. If you have a stress symptom that is frequent and very severe, density is high. Vice versa, Figure 5 shows the advantage of using density as a more accurate parameter to measure psychological burnout.

The COVID-19 pandemic has put educational workers at the center of attention, hence hundreds of thousands of children and families were affected [55,56,57,58]. It also had a heavy burden on the educators and the negative consequences for many who ended up with clear symptoms of psychological burnout, which is one of the consequences causing “wear and tear to the body and the soul”. Burnout has been recently classified by the WHO as a form of work-induced stress [24]. While the WHO does not classify burnout as a medical condition, it is well known to be related to impaired mental health [59] and altered physiological functioning (e.g., increased sympathetic activity) [60]. Importantly, burnout negatively affects the professional functioning of educators as well as the professional services they provide for their students [61,62,63]. Therefore, the prevalence of burnout has often become a monitored metric in many professions. Burnout is usually characterized by three dimensions [45,64,65], namely, the experience of energy depletion or emotional exhaustion, a feeling of reduced personal efficacy with regard to one’s work, and a cynical attitude toward the value of one’s occupation (also termed depersonalization). Of the three dimensions, the first has shown to be the most predictive, given that emotional exhaustion constitutes a core dimension that is clearly linked to pathogenesis and a variety of negative consequences. One important limitation of prior use of burnout is that measuring only frequency does not tell the entire story, hence individual emotions depend not only on it, but also on the severity of each experience. Emotions do not exist independently but interact with each other. Thus, we have recently introduced an approach that enables the modelling of an individual’s emotional life as a dynamic interplay of the frequency and severity of the experiences (stressors) and the capacity to respond to both (i.e., this is the core of the tool called “The Stress Map”, 2021). This, in essence, is the concept of density of emotions. A higher density is indicative of a more solid and consistent parameter to measure emotions given environmental challenges. In this study, albeit with some autocorrelations, a justification is provided to be used in future research; the algorithm of density seems to be very solid both conceptually and psychometrically. Figure 5 shows 3 possible states of density of emotions (psychological burnout: Low, Medium, and High). Each of the latter corresponds, and can be used as a proxy of other correlated consequences. To our knowledge, the association between the density of emotional states and the functioning of the individual was shown in other studies [66,67,68]. Our study shows empirical homogeneity of the Burnout density at the Cronbach alpha of 0.839.

Now, let us turn and discuss the concrete findings of this empirical study and contextualize them. By and large, a comparison of antecedents and profiles of the High vs. Low burnout educators has been summarized in Figure 4. In the paragraphs below, we further discuss these findings.

For the “low burnout-resilient” sample, whether we use frequency, severity, or density, we find that women have suffered more burnout than men, and that the absence of support and control and the composite scale of family sources of stress significantly contribute to increased burnout. The educational sector, for historical reasons, has been dominated by females. Nonetheless, in the past several years, it is also attracting male professionals. There are several studies that show an important gender difference in both the perception of stress as a threat, and also in the manner of these individuals’ responses [62,69]. We have also found a significative relation between gender and burnout in previous studies, in the sense that women report higher scores on the burnout scale than men [61,70]. There might be many reasons for that, but let us propose some. First, in other sectors (such as health) it has been reported that women lack role models, face challenges of dual-career couples, and assume greater responsibility for childbearing. Another possible explanation has to do with the fact that women have a stronger tendency toward certain emotions, such as guilt. Therefore, women are more likely to feel a sense of responsibility to be everything to everyone, and if that feeling is accompanied by a perception of lack of support and control, the psychological consequences are denser. In difficult times, such as what we are experiencing nowadays, the effects are more noticeable. Even before the pandemic, working women in various sectors in the U.S. had been under greater stress and burnout than their male counterparts (68% versus 58%, respectively), and early results during the COVID-19 pandemic, shows a rise to more than 70% [71].

We also find an interactive effect between the composite of work sources of stress and the absence of support and control. Figure 2 shows this interaction effect for low-frequency, low-severity, and low-density of burnout. As it can be observed, in all cases, the perception of not having support or feeling in control make people more “burned out”. These results are consistent with the Karasek [54] model and have been empirically found in a great deal of occupations, including our own studies amongst health professionals in Canada, but also in Spain [70,72]. In the case of educators, numerous studies show the moderating effects of social support on burnout. Russell et al. [73] for example, reported evidence to the stress-moderating role of social support. They state: “teachers who reported that they had supportive supervisors and indicated that they received positive feedback concerning their skills and abilities from others were less vulnerable to burnout”.

Thus, the perception of not having support or feeling in control makes people more burned out, and this increases as family sources increase. However, those with a low perception of absence of support and control are less burned out as family sources increase. A possible explanation for this finding could be that people with the low perception of non-support and control, feel even better as their family sources increase [74]. Maybe because they do not see their family as a source of demand, but rather as one of support. To reinforce this assertion, we can site a study in China among teachers in secondary education. Zhang and Zhu [37], report three major findings: (a) teacher stress causes burnout; (b) work overload is the most common stressor, followed by role conflict and role ambiguity, respectively. Work overload is the only predictor of emotional exhaustion and depersonalization, but none of the stressors are an effective predictor of reduced accomplishment; (c) family and friend support is the most common source of social support, followed by colleague support and supervisor support, respectively, but supervisor support is the most effective in alleviating stress, emotional exhaustion, and reduced accomplishment; family and friend support is the most effective in mitigating depersonalization.

Related to control variables, our results are very consistent with other studies that show significant correlations with the “high burnout” group. Whether we use frequency, severity, or density, we find that people are more burned-out as they get older and that having children to take care of them reduces the severity and density of burnout.

### 4.1. Limitations

Similarly to all studies, ours has limitations that also provide opportunities for future research. A first limitation of the study is its reliance on self-reporting of stress and its antecedents. While self-reporting is prevalent in research on psychological phenomenon, it could be subjective and further limited by the unwillingness of respondents to reveal potentially stigmatizing information. Another limitation is that it could take a long time (about 20–30 min) to finish the survey, leading possible biases and also to non-responses. Although we have checked for the non-response bias, skipping and missing responses leads to a lower sample size for analyses. Finally, our study uses a shortened version of the validated Maslach Burnout Inventory (MBI). The reason is that the questionnaire was too long to guarantee a reasonable response rate and we also had available a proxy shortened measure that was validated earlier. This decision has been reinforced by recent publications, claiming that even a single item burnout measure is psychometrically sound [75]. Other studies, by contrast, propose more caution [76].

### 4.2. Future Lines of Research

For future research in the field of education, we recommend expanding the concept of resilience. There is some evidence that show a significant relation between optimism and lowering the risk of experiencing burnout [9,77]. In the past, educational organizations employed teachers with a high intrinsic engagement and higher commitment to the educational service. Nowadays, the new generation might expect something else from the organizations where they work, including during crisis [78,79]. Teachers are advised to enhance their own resilience by developing a heightened sense of self-efficacy. For schools, work can be conducted to build a resilient school culture by designing a staff development program, building in systems of support, and offering the teachers a greater sense of control over their work.

### 4.3. Practical Implications

This study is expected to help educational administrators and professionals alike in the education sector reflect and have better understanding about the relationship between the way they live and the stress they experience. First, the study shows how work and family interchange and have impact on burnout. Moreover, it shows how depending on the perception of support and control, these sources have a higher or lower impact. On the one hand, those living in the high burnout spectrum with high non-support and control perception feel better when work is more demanding. Therefore, these people take advantage of increased jobs demands. However, for educators in the low burnout category, experiencing similar levels of support and control, are harmed by high family demands. Educational administrators should analyze these perceptions if they want to understand why some people face work demands more stressful than others, under the same conditions. Worth noting that a low perception of non-support and control protects people under high and low burnout situations. Finally, our study shows the vulnerability of women educators in certain situations, namely in the low burnout sample, they are feeling worse than their male colleagues.

## 5. Conclusions

To conclude, despite several limitations, the paper has several important contributions to both the scholarly community and the practitioners occupying the role of teachers. COVID-19 prolonged period was an ideal laboratory to study chronic stress and its impact on mental state (i.e., Burnout). This paper enhances our understanding of the complexity of studying chronic stress and identifying antecedents and profiles of not only the victims (i.e., high burnout group) but also the survivors who are more resilient (the low burnout group). Another original contribution of the paper is the proposal of a robust measure of burnout that is “short and sweet” and measured in units of density [80,81]. By and large, the study shows the strength of using this new (more precise) algorithm to measure psychological burnout. If you have a stress symptom that is frequent and very severe, density is high. [82]. Finally, the study provides a portrait of the combined effect of work stress and family stress and shows the antecedents and profiles connected to the high vs. low burnout groups.

## Figures and Tables

**Figure 1 ijerph-19-00780-f001:**
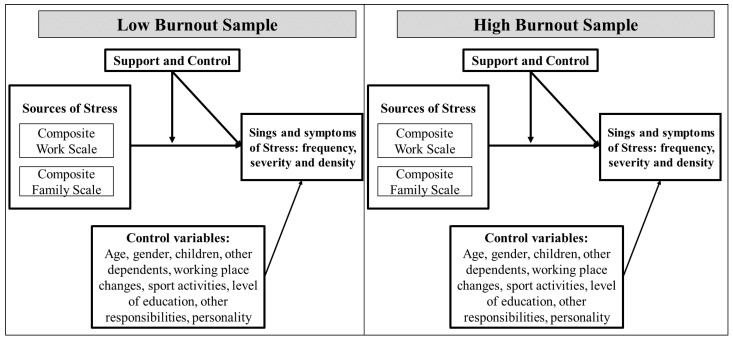
The proposed Conceptual Model.

**Figure 2 ijerph-19-00780-f002:**
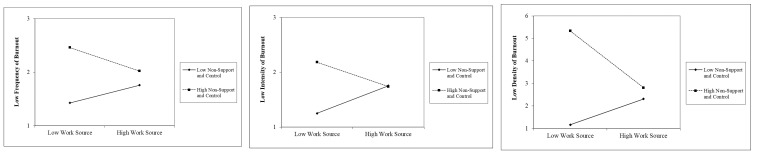
Significant Interactions for Low Frequency, Low Severity and Low Density.

**Figure 3 ijerph-19-00780-f003:**
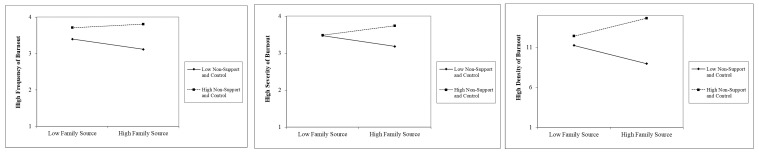
Significant Interactions for High Frequency, High Severity and High Density.

**Figure 4 ijerph-19-00780-f004:**
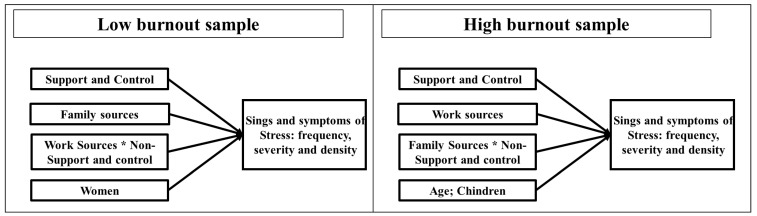
Summary of the significant relations found in the paper. *: multiplied by.

**Figure 5 ijerph-19-00780-f005:**
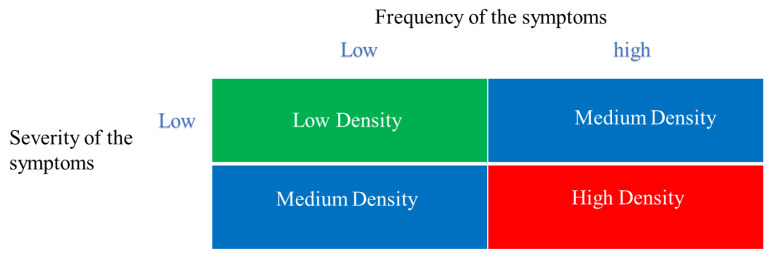
Justification for the use of Density of Burnout in future research.

**Table 1 ijerph-19-00780-t001:** Descriptive statistics and correlations.

		N	Min	Max	Mean	S.D.	1	2	3	4	5	6	7	8	9	10	11	12	13	14	15	16	17	18
1	Age	343	24	67	48.9	9.1	1																	
2	Children	373	0	1	0.7	0.5	0.08	1																
3	Other dependents	373	0	1	0.2	0.4	0.22 **	−0.04	1															
4	Working change	373	0	1	0.5	0.5	−0.05	0.04	0.03	1														
5	Sports activities	373	0	1	0.5	0.5	0.04	−0.03	−0.04	0.00	1													
6	Secondary School	373	0	1	0.3	0.5	−0.01	−0.08	0.03	−0.04	0.01	1												
7	Years 3–5	373	0	1	0.1	0.3	−0.03	0.07	0.10	−0.03	−0.08	−0.26 **	1											
8	Primary School	373	0	1	0.3	0.4	−0.06	0.08	−0.10	0.07	−0.02	−0.41 **	−0.23 **	1										
9	Professional Education	373	0	1	0.1	0.3	−0.05	−0.05	0.04	0.00	0.04	−0.24 **	−0.14 **	−0.21 **	1									
10	Other education years	373	0	1	0.2	0.4	0.15 **	0.00	−0.07	0.00	0.05	−0.31 **	−0.18 **	−0.27 **	−0.16 **	1								
11	Controller	373	1	2	1.2	0.4	0.34 **	0.06	0.05	−0.04	0.05	−0.10	−0.18 **	−0.21 **	−0.12 *	0.61 **	1							
12	Gender	368	0	1	0.6	0.5	−0.11 *	0.01	0.03	0.15 **	−0.11 *	−0.04	0.23 **	0.01	−0.04	−0.13 *	−0.26 **	1						
13	C. Non-S&C	373	1	5	2.1	0.9	−0.10	0.07	−0.02	0.18 **	−0.09	−0.09	0.08	0.09	0.03	−0.11 *	−0.19 **	0.16 **	1					
14	C.F.S	373	1	5	2.5	0.9	−0.05	0.04	−0.03	0.22 **	−0.02	−0.13 *	0.09	0.09	−0.04	0.00	−0.11 *	0.00	0.67 **	1				
15	C.W.S	373	1	5	1.9	0.9	−0.08	0.19 **	0.01	0.15 **	−0.10	−0.05	0.02	0.06	0.01	−0.05	−0.11 *	0.09	0.70 **	0.47 **	1			
16	Type A Personality	373	1	5	2.7	0.8	−0.19 **	−0.02	−0.06	0.05	0.00	−0.11 *	−0.01	0.01	0.11 *	0.04	−0.01	−0.08	0.32 **	0.38 **	0.32 **	1		
17	Burnout-Frequency	373	1	6	3.2	1.3	−0.09	0.04	0.02	0.15 **	−0.16 **	−0.12 *	0.09	0.16 **	−0.01	−0.13 *	−0.18 **	0.22 **	0.67 **	0.51 **	0.47 **	0.29 **	1	
18	Burnout-Severity	373	1	6	3.0	1.3	−0.07	0.04	0.01	0.14 **	−0.15 **	−0.10	0.08	0.17 **	−0.04	−0.12 *	−0.17 **	0.19 **	0.66 **	0.517 **	0.45 **	0.27 **	0.95 **	1

** *p* < 0.01, * *p* < 0.05, see methods for variable definitions; columns 1–18 represent correlations; C. Non-S&C = composite nonsupport and control; C.F.S = composite of family sources of stress; C.W.S = composite of work sources of stress.

**Table 2 ijerph-19-00780-t002:** Regression models for the frequency of stress in low burnout and high burnout samples.

	Low Frequency of Burnout								High Frequency of Burnout							
	Model 0		Model 1		Model 2		Model 3		Model 0		Model 1		Model 2		Model 3	
Control Variables	β	*p*-Value	β	*p*-Value	β	*p*-Value	β	*p*-Value	β	*p*-Value	β	*p*-Value	β	*p*-Value	β	*p*-Value
Age	0.00	0.67	0.00	0.70	0.00	0.62	0.00	0.69	0.01	0.10	0.01	0.15	0.01	0.16	0.01	0.25
Children	0.00	0.99	−0.04	0.70	−0.04	0.70	−0.05	0.67	−0.12	0.37	−0.18	0.14	−0.18	0.15	−0.18	0.16
Other dependents	0.00	0.98	0.09	0.51	0.12	0.40	0.09	0.51	−0.05	0.70	−0.06	0.61	−0.06	0.64	−0.02	0.89
Working change	−0.11	0.30	−0.16	0.12	−0.19	0.08	−0.16	0.12	−0.05	0.68	−0.09	0.38	−0.09	0.38	−0.08	0.42
Sports activities	−0.13	0.23	−0.11	0.28	−0.09	0.39	−0.10	0.33	−0.05	0.63	−0.05	0.63	−0.05	0.64	−0.06	0.55
Secondary School	−0.04	0.83	0.04	0.82	0.08	0.61	0.04	0.79	−0.20	0.43	−0.14	0.52	−0.15	0.51	−0.21	0.36
Years 3–5	0.01	0.97	−0.04	0.86	−0.03	0.88	−0.04	0.85	0.20	0.48	0.23	0.37	0.22	0.39	0.16	0.52
Primary School	0.14	0.45	0.18	0.28	0.21	0.23	0.18	0.29	0.23	0.37	0.23	0.31	0.22	0.33	0.19	0.40
Professional Education	−0.07	0.75	0.03	0.90	0.03	0.89	0.02	0.91	−0.05	0.86	−0.02	0.93	−0.03	0.92	−0.07	0.78
Controller	−0.17	0.29	−0.08	0.62	−0.08	0.60	−0.08	0.62	−0.13	0.63	0.01	0.97	0.01	0.98	−0.02	0.92
Gender	0.17	0.15	**0.23**	**0.05**	**0.22**	**0.06**	**0.23**	**0.05**	0.15	0.23	0.12	0.29	0.12	0.28	0.15	0.20
Type A Personality	0.10	0.21	0.02	0.81	0.03	0.69	0.02	0.83	**0.20**	**0.01**	0.07	0.31	0.07	0.31	0.06	0.37
**Main Variables**																
C. Non-S&C			**0.37**	**0.01**	**0.83**	**0.00**	0.54	0.13			**0.31**	**0.00**	**0.29**	**0.08**	−0.01	0.96
C.W.S			0.00	0.98	**0.49**	**0.09**	−0.01	0.89			0.12	0.10	0.09	0.69	0.10	0.20
C.F.S			**0.16**	**0.07**	0.12	0.21	0.27	0.27			0.00	0.97	0.00	0.97	−0.30	0.10
**Interactions**																
C.W.S * C. Non-S&C					**−0.25**	**0.06**							0.01	0.86		
C.F.S * C. Non-S&C							−0.07	0.63							**0.12**	**0.07**
**Model Fit**																
Adjusted R^2^	−0.01	0.56	0.13	0.00	0.14	0.00	0.13	0.00	0.07	0.02	0.28	0.00	0.28	0.00	0.29	0.00
Increase Adjusted R^2^			0.145	0.00	0.02	0.06	0.00	0.62			0.21	0.00	0.00	0.86	0.01	0.07
F	0.884	0.56	2.615	0.00	2.71	0.00	2.45	0.00	2.13	0.02	5.61	0.00	5.23	0.00	5.54	0.00
Number of Observations	163		163		163		163		177		177		177		177	

See methods for variable definitions. Model 0: only control variables; Model 1: control & independent variables; Model 2: control, independent variables & interaction 1; Model 2: control, independent variables & interaction 2; C. Non-S&C = composite non-support and control; C.F.S = composite of family sources of stress; C.W.S = composite of work sources of stress. *: multiplied by, Bold: significant relationships.

**Table 3 ijerph-19-00780-t003:** Regression models for the severity of stress in low burnout and high burnout samples.

	Low Severity of Burnout								High Severity of Burnout							
	Model 0		Model 1		Model 2		Model 3		Model 0		Model 1		Model 2		Model 3	
Control Variables	β	*p*-Value	β	*p*-Value	β	*p*-Value	β	*p*-Value	β	*p*-Value	β	*p*-Value	β	*p*-Value	β	*p*-Value
Age	0.00	0.91	0.00	0.91	0.00	0.75	0.00	0.91	**0.02**	**0.00**	**0.02**	**0.01**	**0.02**	**0.01**	**0.02**	**0.02**
Children	0.13	0.26	0.05	0.65	0.04	0.71	0.05	0.65	−0.24	0.10	**−0.27**	**0.06**	**−0.26**	**0.06**	−0.26	0.07
Other dependents	0.06	0.67	0.10	0.44	0.11	0.37	0.10	0.44	−0.07	0.63	−0.08	0.56	−0.05	0.72	−0.02	0.90
Working change	0.03	0.80	−0.06	0.56	−0.07	0.45	−0.06	0.56	−0.06	0.60	−0.14	0.25	−0.14	0.24	−0.14	0.25
Sports activities	**−0.19**	**0.06**	**−0.17**	**0.07**	−0.15	0.10	**−0.18**	**0.08**	−0.01	0.91	0.00	0.99	0.00	0.99	−0.01	0.91
Secondary School	−0.03	0.85	0.03	0.86	0.06	0.68	0.02	0.88	−0.19	0.51	−0.07	0.79	−0.12	0.65	−0.17	0.51
Years 3–5	−0.17	0.42	−0.16	0.42	−0.16	0.41	−0.16	0.42	0.07	0.83	0.11	0.70	0.07	0.82	0.01	0.96
Primary School	0.07	0.70	0.12	0.46	0.13	0.43	0.12	0.45	0.14	0.62	0.20	0.46	0.15	0.57	0.13	0.62
Professional Education	−0.17	0.42	−0.06	0.76	−0.06	0.75	−0.06	0.76	−0.33	0.32	−0.24	0.44	−0.27	0.38	−0.30	0.33
Controller	−0.08	0.59	−0.03	0.84	−0.04	0.81	−0.03	0.83	−0.35	0.25	−0.12	0.67	−0.16	0.58	−0.19	0.50
Gender	**0.25**	**0.03**	**0.26**	**0.02**	**0.25**	**0.02**	0.26	0.02	0.21	0.11	0.20	0.12	0.22	0.10	**0.23**	**0.07**
Type A Personality	0.11	0.16	−0.01	0.91	0.01	0.92	−0.01	0.92	0.13	0.12	0.03	0.75	0.03	0.72	0.00	0.96
**Main Variables**																
C. Non-S&C			**0.29**	**0.01**	**0.83**	**0.00**	0.23	0.50			**0.21**	**0.04**	0.05	0.78	−0.26	0.25
C.W.S			0.05	0.64	**0.66**	**0.02**	0.05	0.62			0.08	0.34	−0.16	0.52	0.05	0.52
C.F.S			**0.22**	**0.01**	**0.16**	**0.05**	0.18	0.47			0.08	0.34	0.08	0.34	**−0.37**	**0.08**
**Interactions**																
C.W.S * C. Non-S&C					**−0.30**	**0.02**							0.08	0.30		
C.F.S * C. Non-S&C							0.02	0.86							**0.17**	**0.02**
**Model Fit**																
Adjusted R^2^	0.01	0.29	0.15	0.00	0.17	0.00	0.15	0.00	0.07	0.03	0.20	0.00	0.20	0.00	0.22	0.00
Increase Adjusted R^2^			0.14	0.00	0.03	0.016	0.00	0.86			0.13	0.00	0.00	0.30	0.03	0.02
F	1.19	0.29	3.19	0.00	3.44	0.00	2.97	0.00	1.94	0.03	3.48	0.00	3.33	0.00	3.72	0.00
Number of Observations	187		187		187		187		153		153		153		153	

See methods for variable definitions. Model 0: only control variables; Model 1: control & independent variables; Model 2: control, independent variables & interaction 1; Model 2: control, independent variables & interaction 2; C. Non-S&C = composite non-support and control; C.F.S = composite of family sources of stress; C.W.S = composite of work sources of stress. *: multiplied by, Bold: significant relationships.

**Table 4 ijerph-19-00780-t004:** Regression models for the density of stress (frequency X severity) in low burnout and high burnout samples.

	Low Density of Burnout								High Density of Burnout							
	Model 0		Model 1		Model 2		Model 3		Model 0		Model 1		Model 2		Model 3	
Control Variables	β	*p*-Value	β	*p*-Value	β	*p*-Value	β	*p*-Value	β	*p*-Value	β	*p*-Value	β	*p*-Value	β	*p*-Value
Age	0.01	0.69	0.00	0.91	0.01	0.77	0.00	0.93	**0.18**	**0.00**	**0.15**	**0.01**	**0.15**	**0.01**	**0.14**	**0.02**
Children	0.05	0.92	−0.10	0.81	−0.10	0.80	−0.08	0.85	**−2.32**	**0.05**	**−2.72**	**0.02**	**−2.61**	**0.02**	**−2.50**	**0.02**
Other dependents	0.27	0.63	0.54	0.28	0.63	0.21	0.54	0.29	−0.24	0.85	−0.49	0.66	−0.19	0.87	0.00	0.95
Working change	−0.19	0.66	−0.52	0.19	−0.62	0.11	−0.54	0.18	−0.41	0.69	−0.91	0.34	−0.96	0.31	−0.72	0.38
Sports activities	−0.56	0.19	−0.44	0.25	−0.36	0.35	−0.48	0.22	−0.31	0.75	−0.31	0.72	−0.28	0.75	−0.26	0.62
Secondary School	0.20	0.76	0.54	0.37	0.73	0.22	0.51	0.40	−1.66	0.45	−1.05	0.60	−1.43	0.48	−1.85	0.36
Years 3–5	−0.14	0.87	−0.23	0.77	−0.20	0.79	−0.22	0.78	0.98	0.70	1.23	0.59	0.85	0.71	0.44	0.84
Primary School	0.56	0.44	0.87	0.19	0.94	0.15	0.90	0.18	1.46	0.52	1.53	0.45	1.23	0.55	0.91	0.59
Professional Education	−0.32	0.70	0.09	0.91	0.11	0.88	0.10	0.90	−2.06	0.44	−1.58	0.51	−1.79	0.46	−2.25	0.36
Controller	−0.72	0.26	−0.22	0.71	−0.23	0.69	−0.22	0.71	−3.32	0.16	−1.79	0.40	−1.94	0.36	−2.12	0.30
Gender	0.61	0.19	**0.92**	**0.04**	**0.88**	**0.04**	**0.92**	**0.04**	1.27	0.26	1.16	0.26	1.31	0.21	1.42	0.16
Type A Personality	0.43	0.17	0.07	0.80	0.13	0.65	0.09	0.78	**1.76**	**0.01**	0.54	0.38	0.57	0.36	0.54	0.44
**Main Variables**																
C. Non-S&C			**1.41**	**0.01**	**3.52**	**0.00**	0.61	0.66			**2.30**	**0.00**	0.89	0.55	2.47	0.38
C.W.S			−0.24	0.55	**2.09**	**0.06**	−0.17	0.69			0.89	0.19	−1.21	0.54	−2.73	0.36
C.F.S			**0.94**	**0.00**	**0.72**	**0.03**	0.38	0.69			0.66	0.32	0.63	0.34	**−1.90**	**0.07**
**Interactions**																
C.W.S * C. Non-S&C					**−1.18**	**0.02**							0.69	0.25		
C.F.S * C. Non-S&C							0.33	0.53							**1.13**	**0.02**
**Model Fit**																
Adjusted R^2^	−0.01	0.65	0.16	0.00	0.19	0.00	0.16	0.00	0.09	0.01	0.27	0.00	0.27	0.00	0.29	0.00
Increase Adjusted R^2^			0.18	0.00	0.03	0.02	0.00	0.53			0.18	0.00	0.01		0.02	0.25
F	0.8	0.65	3.15	0.00	3.36	0.00	2.97	0.00	2.43	0.01	5.28	0.00	5.04	0.00	5.38	0.00
Number of Observations	166		166		166		166		174		174		174		174	

See methods for variable definitions. Model 0: only control variables; Model 1: control & independent variables; Model 2: control, independent variables & interaction 1; Model 2: control, independent variables & interaction 2; C. Non-S&C = composite non-support and control; C.F.S = composite of family sources of stress; C.W.S = composite of work sources of stress. *: multiplied by, Bold: significant relationships.

## Data Availability

Not applicable.

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
