# Peer review of "Exploring High vs. Low Burnout amongst Public Sector Educators: COVID-19 Antecedents and Profiles"

_ijerph, 2022, doi:10.3390/ijerph19020780_

Round 1

Reviewer 1 Report

I would recommend the authors to point out limitations of the research and indicate lines of research

Author Response

EXPLORING HIGH VS LOW BURNOUT AMONGST PUBLIC SECTOR EDUCATORS:

COVID-19 ANTECEDENTS AND PROFILES

Response to Reviewer #1

First, let us thank you for your constructive comments. Below, each of your comments to the previous version is repeated in conjunction with our responses. Your comments are bold faced.

I would recommend the authors to point out limitations of the research and indicate lines of research.

Thank you for your comment. We have now a limitations, future line of research an practical implications sections.

We very much appreciate your comments. In following your recommendations, we were able to substantially improve the quality of the manuscript. Thanks.

Reviewer 2 Report

Although the topic is very interesting and contemporary, there are some serious flaws, which should be solved.

  1. Introduction – pretty long, should be summarized (it is written more as a thesis, and even though it is very beautifully written, it still needs to be shorten)
  2. Brief Literature Review – pretty long, it would be advisable to shorten

Lines 109 – 110, and 117-118: Parentheses not closed (i.e., one ) is missing)

Figure 1 – Unclear and imprecise term “student’s year” – was is meant: ”level of education”? It is more official term and has clearer meaning

Line 177 - Unclear meaning of “resources” in the context of sentence. Was it meant “job demands”? I think that this last sentence is quite confusing and probably unnecessary.

Line 102-193 – “socioprofessionals as: level of education service (secondary school, primary school, years 3 to 5, professional education and other education years)”, should be better defined (it is a bit confusing like this). Probably, this should be defined: “socioprofessionals as: level of education (primary school, secondary school, post-secondary education but not tertiary, tertiary education, and post-tertiary education). The order of level of education should increase both in the text and in the tables (e.g., first should be primary education, the last post-tertiary education). It is not clear why just one scale is not given for the level of education in the analysis (e.g., scale from 1-5, with: 1 for primary, and 5 for post-tertiary education) (similarly as for  C.F.S, C.W.S, C. Non-S&C scales). I would suggest regression analysis with educational level as a scale. This way is also acceptable, since it shows that burnout was the highest in primary education group, and the lowest in post-tertiary education group. I suppose that educational level (as a scale) would be a negative predictor in the regression analysis.

  1. Methods and Results

Line 196 – Was the questionnaire sent to all of the 107,269 educators, to ask them to participate in the survey? This should be defined. If not, was there any calculation on the minimal sample size? Or were there any post-hoc analyses of the study sample number representativeness? Were there any ethical approvals for the study?

Lines 210-305 – I would suggest placing quotation marks for all the questions included in the questionnaire, e.g., (1) “I felt that I had too many things to accomplish at work”, which measures work overload; (2) “I felt that I am being underpaid for the type of work I do”, which measures pay inequity;..

Lines 211-222 – It is not clear why the authors shortened the validated Maslach Burnout Inventory (MBI), which is the “gold standard” for burn-out assessment. They said: ”to shorten the overall length of the survey”, but was it really justified? From 22 item, they shorted it to only 4 items. Was this new scale from only 4 items really representative for the burnout assessment? The authors of MBI declare that the shorten versions are not reliable enough https://www.mindgarden.com/316-mbi-educators-survey#horizontalTab4 (see later). If the burnout was the main outcome of this study, maybe it was better to keep the original questionnaire. At least, this should be pointed-out in discussion, particularly in “the study limitations”.

The scoring system also should be presented, e.g., it was “Never (0); A few times a year or less (1); Once a month or less (2); A few times a month (3); Once a week (4); A few times a week (5); Every day (6)”, for the frequency scale (i.e., dimension). For the intensity scale, what was the scoring system? Please note – The intensity scale, present in the original first edition of MBI, is in the later editions removed, because of redundancy of intensity and frequency ratings. Therefore, in accordance with 4th edition, only the frequency scale should be used.

What was the maximum score in the 4-item MBI scale? 24? This should be given, also.

Line 224 – In the last MBI edition, only “burnout frequency” is used, “burnout severity” is excluded, because of high correlation between them. Please, give the scoring system in methodology for both frequency and intensity (severity) scores, with maximum scores.  Why “burnout density” was not presented in the Table 1?

Line 228 – How the groups were separated? By the mean MBI and by median? This should be given in the methodology.

Lines 241-278 – What were the scoring systems for the answers in the scales (0-6, as in the MBI, or 0-1, yes-no)? This should be specified. What was the maximum for each scales? 7, 5 and 10, respectively? This also should be added for each scale

Lines 265-278 – Why the lack of support and lack of control were united in one scale, when they are totally separated items, with different, separate impact on burnout, and should be analyzed and discussed separately. I would suggest re-analysis of regression analysis, with 2 separate scales (for support and control), and separate discussion. Please, give the scoring system and maximum of the scale(s).

Lines 294-295 – see above how to better define

Lines 299-304 – what were scoring system and maximum of the scale?

I think that in the text gender should be on the first place, together with age, and later the other control variables (small rearrangements of text and tables suggested).

Also Statistics sub-section should be presented in the Methods section, with definition of Pearson correlation (Table 1),  linear regression analysis –(OLS), (which method – stepwise?), dependent and controlling/dummy variables, test of VIFs for multicolinearity, etc. Why Man-Whitney test and Chi-square were not used for comparison between low- and high- burnout groups? Describe also SUR here.

4. Results - should be section 4, not 3

In general, with the way the results are presented is quite difficult to follow and to draw the main results.

Table 1 – for nominal data N and % should be given in the column, with explanation (no-0, yes-1, male – 0, female - 1).

Actually, maybe an additional introductory table should be presented with the study sample characteristics (mean, SD, min, max – for numeric data and N and % for nominal data). Also give the numbers of the low-burnout and high-burnout group, with the difference between them (Man Whitney and Chi-square).

Table 1 – what is 1-18 in the columns? Please,  specify in the table footnote that those are the same numbers as the serial numbers of the explored items in the rows.

Table 1 – Needed definitions for all abbreviations (C.W.S, C.F.S, C. Non-S&C). Actually, I would suggest to avoid those abbreviations in the table and to give simply “Work sources of Stress”, “Family sources of Stress, ”Lack of support and control”, it is much more observable and easier for a reader. Maybe also in the text I would suggest to avoid these abbreviations, because are not necessary and makes it more difficult for a reader to follow.

Also Pearson’s coefficients of correlations should be declared in the correlation matrix Table 1 (in the footnote).

Table 1 – why burnout density was not shown?

Table 1 – the maximum score on MBI frequency is 24, why the values are given as 1-6? Should the sum of all 4 item be given? This should be explained.

Table 1 – all numbers should be with uniform decimals (2 or 3 decimals), except for N, min, max (e.g.,  0.52, not 0.517). Numbers should be uniformly presented (0.01 or .01)

There should be a short textual explanation what is found to correlate with the burnout. It is not easy to follow from the Table 1 (since the columns were not clearly labeled), so would be useful to draw a small conclusion from this analysis.

For example, “Burnout correlated with all of the examined variables, except with age, having children and other dependents, and some education levels. Having only primary education was positively associated with the job burnout, while having post-tertiary education was negatively associated with burnout. Also having a controller function at work was negatively associated with burnout, as well as having sport activities. Female gender was also positively associated with burnout, as well as type A personality. Job burnout was positively associated with the scores on “Work sources of stress”, “Family sources of stress, and ”Lack of support and control” scales, with the last having the highest level of association (Pearson’s coefficient of correlation 0.66).”

Lines 1-15 (after Table 1) – this should go to Statistics subsection of Methods

Table 2 – define all abbreviations and models in footnotes. (What were the models 0-3? What is the difference between them? )

Lines 1-26 (After Figure 2) and Tables 2-4 , Figure 2:

I think it is quite confusing to present all of these models (for frequency, severity and density). Particularly because it is not quite clear what is the difference in frequency and severity calculation (it was not explained in the methods). The authors should stick to just one of the models in the article, and if necessary – the others give as a supplementary material. It is already complicated to present separately low-burnout and high-but group. (The last edition of MBI only contains the frequency scale, and it will be good to stick only with this scale, but if the main point is on burnout density, maybe to choose to present only these results)

Why there is no regression analysis for the whole group, without low-burnout and high-but group separation?  Instead of giving frequency, intensity and density dimensions (which are all similar and inter-correlated), it would be much more useful to give regression analysis for the whole set of participants. In my opinion, making separate regressions in low-burnout and high-burnout groups is probably unnecessary, and does not give any additional meaningful and significant information. Maybe is better to compare those 2 groups (see below).

There are not given the numbers of participants in each burnout group. Where those numbers enough large to make reliable regression analyses?

Why there are not shown differences between the 2 burnout groups in all main variables and co-variables, by Man-Whitney and Chi square test? This will be very useful.

There are not explanations what were the models 0-3, and which of them was the best.

Lines 1-26 (After Figure 2) – The text is pretty confusing to follow.

I think lack of support and lack of control should be separated, as 2 separate scales, and the analyses should be repeated. What is more important for burnout: lack of support or lack of control?

5. Discussion:

As the authors themselves pointed-out, it is difficult to interpret all of these results presented here in such way. I would suggest to use only one scale (e.g., frequency), because only it is in the last edition of MBI (i.e., only in the first edition the intensity/severity scale was used).

However, if already the original and validated 22- item MBI for educators was not used (the shorten and not validated 4-item MBI is used), and the authors like to apply “the density” scale, they could give that “density approach”, even though this is not validated. In that case, in results only density table and graph should be given, and the other 2 could be placed in supplementary material.

Some parts of discussion were not related to the results presented in this study (e.g., 3 different  burnout dimensions: emotional exhaustion, personal  accomplishment and depersonalization (cynicism) were not properly covered by the 4-item questionnaire, and therefore should not be discussed here, because the authors did not give the separate results for these dimensions).

In my opinion, making discussion on predictors of burnout and high-burnout groups is unnecessary, and does not give any meaningful and significant information. Particularly in the low-burnout group, it is without significant point to discuss the predictors of burnout. Much better would be to compare the 2 groups in sex, age, level of education and all other parameters… , and give the regression analysis on the whole group. This would be more useful.

There is very little in results and discussion on COVID-19, particularly on the burden on work- change (working on- line, from home, on on-line platforms, which is a novelty for majority of teachers). This aspect was mentioned in introduction, but is completely lost later on.

In discussion, more comparison with other recent studies on burnout among teachers during COVID-19 should be given (there are many published articles on that topic, in other countries, and comparison could be made).

In study limitations it was not mentioned that 4-item MBI was used, instead of 22-item, so not validated questionnaire for burnout was not used. The official cite for MBI https://www.mindgarden.com/316-mbi-educators-survey#horizontalTab4 gives the limitations of this shorten versions:

  • Using only a few MBI items produces greater measurement error, reducing confidence in burnout findings.
  • Using only a few MBI items reduces the sampling of the burnout construct.
  • Using only a few MBI items reduces reliability. Inferences about results should only be drawn from a large number of people, and individuals should never be assessed with just a few items.
  • Each MBI item provides only 6 intervals between scores, whereas a five-item scale provides 30 intervals.
  • Results from studies using only a few MBI items may lack validity and reliability.
  • Using only a few MBI items does not permit profile analysis. Profiles are the result of extensive research on MBI scales. The profiles are predictive and descriptive, and they provide additional meaning.
  • Using only a few MBI items may not adequately measure the three MBI scales.
  • Using only a few MBI items is not consistent with the ICD-11 inclusion of burnout which includes all three constructs.
  • Using only a few MBI items precludes normative comparisons with previous research.
  • Using only a few MBI items precludes normative comparisons with data in the MBI Manual.
  • Defining a high level of burnout is problematic when using only a few MBI items.
  • As with the MBI scales, the scores of only a few MBI items cannot be combined to produce a single composite burnout score.
  • In situations where minimizing the MBI item count is important, researchers are encouraged to consider using the MBI General Survey (16 items).
  • Even though fewer MBI items might correlate well with the entire MBI, the shorter form can create high rates of false-positives. See “The Abbreviated Maslach Burnout Inventory Can Overestimate Burnout: A Study of Anesthesiology Residents.” Journal of Clinical Medicine 2020, 9, 61.

6. Conclusion:

There is no a separate section covering conclusion

7. Literature:

There are many old references, including the Maslach MBI.  The 4th, last edition of Maslach should be used.

In my opinion, the article should be completely rewritten, particularly methods and results section, with additional statistical analyses (please, see above). It should be generally shortened and simplified. 

Author Response

Ms. Ref. No.: ijerph-1504738

Title: EXPLORING HIGH VS LOW BURNOUT AMONGST PUBLIC SECTOR EDUCATORS:

COVID-19 ANTECEDENTS AND PROFILES

Response to Reviewer #2

First, let us thank you for your constructive comments. Below, each of your comments to the previous version is repeated in conjunction with our responses. Your comments are bold faced.

Although the topic is very interesting and contemporary, there are some serious flaws, which should be solved.

Thank you for your encouragement! We have dealt with all of your comments. We can only hope to have improved the paper. Thanks.

  1. Introduction – pretty long, should be summarized (it is written more as a thesis, and even though it is very beautifully written, it still needs to be shorten)

Thank you for your comment. We have now revised the introduction section to summarize it. We can only hope that you like it as it is now.

  1. We Brief Literature Review – pretty long, it would be advisable to shorten

Thank you for your comment. We have now revised the literature review section to summarize it. We can only hope that you like it as it is now.

Lines 109 – 110, and 117-118: Parentheses not closed (i.e., one ) is missing)

Figure 1 – Unclear and imprecise term “student’s year” – was is meant: ”level of education”? It is more official term and has clearer meaning

Line 177 - Unclear meaning of “resources” in the context of sentence. Was it meant “job demands”? I think that this last sentence is quite confusing and probably unnecessary.

Line 102-193 – “socioprofessionals as: level of education service (secondary school, primary school, years 3 to 5, professional education and other education years)”, should be better defined (it is a bit confusing like this). Probably, this should be defined: “socioprofessionals as: level of education (primary school, secondary school, post-secondary education but not tertiary, tertiary education, and post-tertiary education). The order of level of education should increase both in the text and in the tables (e.g., first should be primary education, the last post-tertiary education). It is not clear why just one scale is not given for the level of education in the analysis (e.g., scale from 1-5, with: 1 for primary, and 5 for post-tertiary education) (similarly as for  C.F.S, C.W.S, C. Non-S&C scales). I would suggest regression analysis with educational level as a scale. This way is also acceptable, since it shows that burnout was the highest in primary education group, and the lowest in post-tertiary education group. I suppose that educational level (as a scale) would be a negative predictor in the regression analysis.

Thank you very much for all of these comments. We have corrected all of them in the new version of the paper. In reference to your comment, “It is not clear why just one scale is not given for the level of education in the analysis (e.g., scale from 1-5, with: 1 for primary, and 5 for post-tertiary education) (similarly as for  C.F.S, C.W.S, C. Non-S&C scales)”, we believe that giving the different options provide a richer explanation of your finings. We can only hope that you agree with it.

  1. Methods and Results

Line 196 – Was the questionnaire sent to all of the 107,269 educators, to ask them to participate in the survey? This should be defined. If not, was there any calculation on the minimal sample size? Or were there any post-hoc analyses of the study sample number representativeness? Were there any ethical approvals for the study?

Thank you very much for your comment. We have now better explained the sample characteristics in the methods section as follows:We surveyed personnel working in the education system in Andalusia. There were 107,269 educators at the time of data collection. From them, about 67% are women. We sent the questionnaire to all of the population. Out of these, 340 questionnaires were completed and represent the sample for which all analyses were affected. We tested the non-response bias and found no statistically significant differences for age, gender, and marital status. Even more, our sample is representative of the population in terms of proportion of men and men, 64% are women in our sample and 65% in the full population.”

Lines 210-305 – I would suggest placing quotation marks for all the questions included in the questionnaire, e.g., (1) “I felt that I had too many things to accomplish at work”, which measures work overload; (2) “I felt that I am being underpaid for the type of work I do”, which measures pay inequity;..

Again, thank you very much for your comment. We have now placed quotation marks in all of the questions included in the questionnaire.

Lines 211-222 – It is not clear why the authors shortened the validated Maslach Burnout Inventory (MBI), which is the “gold standard” for burn-out assessment. They said: ”to shorten the overall length of the survey”, but was it really justified? From 22 item, they shorted it to only 4 items. Was this new scale from only 4 items really representative for the burnout assessment? The authors of MBI declare that the shorten versions are not reliable enough https://www.mindgarden.com/316-mbi-educators-survey#horizontalTab4 (see later). If the burnout was the main outcome of this study, maybe it was better to keep the original questionnaire. At least, this should be pointed-out in discussion, particularly in “the study limitations”.

Again, thank you very much for your comment. We have now included this limitation in the limitations section as follows: “Finally, our study uses a shortened version of the validated Maslach Burnout Inventory (MBI). The reason is that the questionnaire was too long to guarantee a reasonable re-sponse rate. While, based on recent publications, even a single item burnout measure is psychometrically sound [102], other studies propose that caution should be taken [103].

The scoring system also should be presented, e.g., it was “Never (0); A few times a year or less (1); Once a month or less (2); A few times a month (3); Once a week (4); A few times a week (5); Every day (6)”, for the frequency scale (i.e., dimension). For the intensity scale, what was the scoring system? Please note – The intensity scale, present in the original first edition of MBI, is in the later editions removed, because of redundancy of intensity and frequency ratings. Therefore, in accordance with 4th edition, only the frequency scale should be used. What was the maximum score in the 4-item MBI scale? 24? This should be given, also. Line 224 – In the last MBI edition, only “burnout frequency” is used, “burnout severity” is excluded, because of high correlation between them. Please, give the scoring system in methodology for both frequency and intensity (severity) scores, with maximum scores.  Why “burnout density” was not presented in the Table 1?

Again, thank you very much for your comment. We have now presented the scoring system. Even more, since there is a discussion about intensity and frequency, we wanted to include both, and a new one “density”, to clarify previous studies and contribute to the literature.

Again, thank you very much for your comment. All minimum and maximum scores are now in Table 1. In relation to MBI, it is 6 (the mean value of 24 divided by 4). The reason why density is not in table 1 is because it is just Frequency X intensity. Therefore, the maximum is 36 and the mean is proportional to the mean of frequency and intensity.

Line 228 – How the groups were separated? By the mean MBI and by median? This should be given in the methodology.

Again, thank you very much for your comment. We have included all of this information in page 9 of the new version of the paper as follows: “The threshold to distinguish between low (i.e., resilient people hereafter) and high burnout, is based on the mean value. The results obtained with the mean values are similar to those obtained with the median values (not presented here due to space limitations).”

Lines 241-278 – What were the scoring systems for the answers in the scales (0-6, as in the MBI, or 0-1, yes-no)? This should be specified. What was the maximum for each scales? 7, 5 and 10, respectively? This also should be added for each scale

Again, thank you very much for your comment. Table 1 of the new version of the paper includes now all of minimum and maximum values. Even more, following your previous recommendations, we now explain the scoring system in the methodology of the paper for each scale.

Lines 265-278 – Why the lack of support and lack of control were united in one scale, when they are totally separated items, with different, separate impact on burnout, and should be analyzed and discussed separately. I would suggest re-analysis of regression analysis, with 2 separate scales (for support and control), and separate discussion. Please, give the scoring system and maximum of the scale(s).

Again, thank you very much for your comment. We agree with the reviewer. However, conducted factor analysis and lack of support and lack of control items were highly correlated. Statistically, it was impossible to compute them as independent variables. Again, all of the scoring systems are now in the manuscript.

Lines 294-295 – see above how to better define

Again, thank you very much for your comment. We have used your comment for a better definition.

Lines 299-304 – what were scoring system and maximum of the scale?

Again, thank you very much for your comment. All of the scoring systems are now in the manuscript.

I think that in the text gender should be on the first place, together with age, and later the other control variables (small rearrangements of text and tables suggested).

Again, thank you very much for your comment. We have followed the reviewer advises for a better flow of the paper.

Also Statistics sub-section should be presented in the Methods section, with definition of Pearson correlation (Table 1),  linear regression analysis –(OLS), (which method – stepwise?), dependent and controlling/dummy variables, test of VIFs for multicolinearity, etc. Why Man-Whitney test and Chi-square were not used for comparison between low- and high- burnout groups? Describe also SUR here.

Again, thank you very much for your comment. We have followed the reviewer advises an we have now a correlation table (1), OLS with stepwise, and explained that all VIF were below 4. The description of SUR and references about it are also in the manuscript.

  1. Results - should be section 4, not 3

Again, thank you very much for your comment. We have a single section 3 which is called methods and results.

In general, with the way the results are presented is quite difficult to follow and to draw the main results.

Actually, maybe an additional introductory table should be presented with the study sample characteristics (mean, SD, min, max – for numeric data and N and % for nominal data). Also give the numbers of the low-burnout and high-burnout group, with the difference between them (Man Whitney and Chi-square).

Again, thank you very much for your comment. We have followed the reviewer advises for a better flow of the paper.

Table 1 – what is 1-18 in the columns? Please,  specify in the table footnote that those are the same numbers as the serial numbers of the explored items in the rows.

Again, thank you very much for your comment. We have now added a footnote explaining that columns 1-18 represent correlations.

Table 1 – Needed definitions for all abbreviations (C.W.S, C.F.S, C. Non-S&C). Actually, I would suggest to avoid those abbreviations in the table and to give simply “Work sources of Stress”, “Family sources of Stress, ”Lack of support and control”, it is much more observable and easier for a reader. Maybe also in the text I would suggest to avoid these abbreviations, because are not necessary and makes it more difficult for a reader to follow.

Again, thank you very much for your comment. All of abbreviations are now explained in the methods section.

Also Pearson’s coefficients of correlations should be declared in the correlation matrix Table 1 (in the footnote).

Again, thank you very much for your comment. We have now added a footnote explaining that columns 1-18 represent correlations.

Table 1 – why burnout density was not shown?

Again, thank you very much for your comment. As we have already explained you before, the reason why density is not in table 1 is because it is just Frequency X intensity. Therefore, the maximum is 36 and the mean is proportional to the mean of frequency and intensity.

Table 1 – the maximum score on MBI frequency is 24, why the values are given as 1-6? Should the sum of all 4 item be given? This should be explained.

Again, thank you very much for your comment. As we have explained you before, the maximum is 6 (24/4)

Table 1 – all numbers should be with uniform decimals (2 or 3 decimals), except for N, min, max (e.g.,  0.52, not 0.517). Numbers should be uniformly presented (0.01 or .01)

Again, thank you very much for your comment. We have followed the journal advices about the number of decimal.s

There should be a short textual explanation what is found to correlate with the burnout. It is not easy to follow from the Table 1 (since the columns were not clearly labeled), so would be useful to draw a small conclusion from this analysis. For example, “Burnout correlated with all of the examined variables, except with age, having children and other dependents, and some education levels. Having only primary education was positively associated with the job burnout, while having post-tertiary education was negatively associated with burnout. Also having a controller function at work was negatively associated with burnout, as well as having sport activities. Female gender was also positively associated with burnout, as well as type A personality. Job burnout was positively associated with the scores on “Work sources of stress”, “Family sources of stress, and ”Lack of support and control” scales, with the last having the highest level of association (Pearson’s coefficient of correlation 0.66).”

Again, thank you very much for your comment. We have now added textual explanation about what is found to correlate with burnout. The reason why is was not there before is that we had space constrictions.

Lines 1-15 (after Table 1) – this should go to Statistics subsection of Methods

Again, thank you very much for your comment. It is now in the methods and results section.

Table 2 – define all abbreviations and models in footnotes. (What were the models 0-3? What is the difference between them? )

Again, thank you very much for your comment. All of the abbreviations are explained in the methods and results sections. Models 0-3 are now explained as a foot note in tables 2-4.

Lines 1-26 (After Figure 2) and Tables 2-4 , Figure 2:

I think it is quite confusing to present all of these models (for frequency, severity and density). Particularly because it is not quite clear what is the difference in frequency and severity calculation (it was not explained in the methods). The authors should stick to just one of the models in the article, and if necessary – the others give as a supplementary material. It is already complicated to present separately low-burnout and high-but group. (The last edition of MBI only contains the frequency scale, and it will be good to stick only with this scale, but if the main point is on burnout density, maybe to choose to present only these results)

Again, thank you very much for your comment. We have now clarified the calculation for frequency, severity, and density. As we have explained before, one of the contributions of the paper is to present the 3 dependent variables.

Why there is no regression analysis for the whole group, without low-burnout and high-but group separation?  Instead of giving frequency, intensity and density dimensions (which are all similar and inter-correlated), it would be much more useful to give regression analysis for the whole set of participants. In my opinion, making separate regressions in low-burnout and high-burnout groups is probably unnecessary, and does not give any additional meaningful and significant information. Maybe is better to compare those 2 groups (see below).

Again, thank you very much for your comment. Given space limitations, we could present the full sample as supplement material. Having two groups is also a contribution of this paper.

There are not given the numbers of participants in each burnout group. Where those numbers enough large to make reliable regression analyses?

Again, thank you very much for your comment. We have now included this information as follows: “Data was primarily analyzed using an ordinary least square (OLS) procedure. In addition, in order to examine possible differences in patterns amongst high and low burned-out people, we use seemingly unrelated regression (SUR) with separate re-gression equations for each subsample (see Figures 1, 3 and Tables 2–4). The threshold to distinguish between low (i.e., resilient people hereafter) and high burnout, is based on the mean value. The results obtained with the mean values are similar to those obtained with the median values (not presented here due to space limitations). Specifically, we ran three sets of regression equations for the subsamples. The three sets contain two regression equations. In the first case, one regression was based on the observations of those who showed low frequency of burnout (164 individuals), and the other was based on the observations of those who showed high frequency of burnout (176 individuals). The second set contains observations of those who showed low severity of burnout (164 individuals), and the other was based on the observations of those who showed high severity of burnout (176 individuals). Finally, the third set contains observations of those who showed low density of burnout (164 individuals), and the other was based on the observations of those who showed high density of burnout (176 individuals)”.

There are not explanations what were the models 0-3, and which of them was the best.

Again, thank you very much for your comment. We have now explained what 0-3 models explain and using R2, we have explained which is the best.

Lines 1-26 (After Figure 2) – The text is pretty confusing to follow.

Again, thank you very much for your comment. We have reviewed this confusing test.

I think lack of support and lack of control should be separated, as 2 separate scales, and the analyses should be repeated. What is more important for burnout: lack of support or lack of control?

Again, thank you very much for your comment. As we have explained before, it is statistically impossible to separate these variables.

  1. Discussion:

As the authors themselves pointed-out, it is difficult to interpret all of these results presented here in such way. I would suggest to use only one scale (e.g., frequency), because only it is in the last edition of MBI (i.e., only in the first edition the intensity/severity scale was used). However, if already the original and validated 22- item MBI for educators was not used (the shorten and not validated 4-item MBI is used), and the authors like to apply “the density” scale, they could give that “density approach”, even though this is not validated. In that case, in results only density table and graph should be given, and the other 2 could be placed in supplementary material.

Again, thank you very much for your comment. We have already explained why we use the different Dependent Variables.

Some parts of discussion were not related to the results presented in this study (e.g., 3 different  burnout dimensions: emotional exhaustion, personal  accomplishment and depersonalization (cynicism) were not properly covered by the 4-item questionnaire, and therefore should not be discussed here, because the authors did not give the separate results for these dimensions).

Again, thank you very much for your comment. We have eliminate all of non-use material from the discussion.

In my opinion, making discussion on predictors of burnout and high-burnout groups is unnecessary, and does not give any meaningful and significant information. Particularly in the low-burnout group, it is without significant point to discuss the predictors of burnout. Much better would be to compare the 2 groups in sex, age, level of education and all other parameters… , and give the regression analysis on the whole group. This would be more useful.

Again, thank you very much for your comment. We’ll try to develop future research based on your suggestions.

There is very little in results and discussion on COVID-19, particularly on the burden on work- change (working on- line, from home, on on-line platforms, which is a novelty for majority of teachers). This aspect was mentioned in introduction, but is completely lost later on. In discussion, more comparison with other recent studies on burnout among teachers during COVID-19 should be given (there are many published articles on that topic, in other countries, and comparison could be made).

Again, thank you very much for your comment. We have now talked a bit more about Covid in the discussion. For example, we now say: “In difficult times, like what we are experiencing nowadays, the effects are more no-ticeable. Even before the pandemic, working women in various sectors in the U.S. had been under greater stress and burnout than their male counterparts (68% versus 58%), and early results during the COVID-19 pandemic, shows a rise to more than 70% [90].”

In study limitations it was not mentioned that 4-item MBI was used, instead of 22-item, so not validated questionnaire for burnout was not used. The official cite for MBI https://www.mindgarden.com/316-mbi-educators-survey#horizontal

Tab4 gives the limitations of this shorten versions:

  • Using only a few MBI items produces greater measurement error, reducing confidence in burnout findings.
  • Using only a few MBI items reduces the sampling of the burnout construct.
  • Using only a few MBI items reduces reliability. Inferences about results should only be drawn from a large number of people, and individuals should never be assessed with just a few items.
  • Each MBI item provides only 6 intervals between scores, whereas a five-item scale provides 30 intervals.
  • Results from studies using only a few MBI items may lack validity and reliability.
  • Using only a few MBI items does not permit profile analysis. Profiles are the result of extensive research on MBI scales. The profiles are predictive and descriptive, and they provide additional meaning.
  • Using only a few MBI items may not adequately measure the three MBI scales.
  • Using only a few MBI items is not consistent with the ICD-11 inclusion of burnout which includes all three constructs.
  • Using only a few MBI items precludes normative comparisons with previous research.
  • Using only a few MBI items precludes normative comparisons with data in the MBI Manual.
  • Defining a high level of burnout is problematic when using only a few MBI items.
  • As with the MBI scales, the scores of only a few MBI items cannot be combined to produce a single composite burnout score.
  • In situations where minimizing the MBI item count is important, researchers are encouraged to consider using the MBI General Survey (16 items).
  • Even though fewer MBI items might correlate well with the entire MBI, the shorter form can create high rates of false-positives. See “The Abbreviated Maslach Burnout Inventory Can Overestimate Burnout: A Study of Anesthesiology Residents.” Journal of Clinical Medicine 2020, 9, 61.

Again, thank you very much for your comment. We have now introduced this limitation and presented other recent papers in relation to the scale.

  1. Conclusion:

There is no a separate section covering conclusion

Again, thank you very much for your comment. There is a conclusion section now.

  1. Literature:

There are many old references, including the Maslach MBI.  The 4th, last edition of Maslach should be used.

Again, thank you very much for your comment. We have now included new references in relation to the Maslach MBI.

In my opinion, the article should be completely rewritten, particularly methods and results section, with additional statistical analyses (please, see above). It should be generally shortened and simplified. 

Again, thank you very much for your comment. We can only hope that all of our changes have improved the manuscript.

We very much appreciate your comments. We have rewritten much of the sections. In following your recommendations, we were able to substantially improve the quality of the manuscript. Thanks.

Reviewer 3 Report

I have read this manuscript on burnout in public sector educators with interest. Overall, it is well written, fits well to the special issue, and makes a meaningful contribution to the literature. There are, however, a few issues that should be considered before being able to accept this paper for publication.

  1. Abstract: Should contain information about the sample size and the analyses.
  2. Generally, I wondered whether the distinction burnout VS “those that were able to develop resilience” fits well to (a) what is theoretically feasible (not ending up in burnout doesn’t necessary mean having developed resilience) and (b) what was empirically investigated (high vs low burnout levels).
  3. The introduction is a bit lengthy. I would find it helpful to already situate the present study and the research aims in the very first paragraph. Doing so, would help the readers to make more sense of the following.
  4. Multiple studies have already been conducted on stress experiences and burnout during COVID19. In particular, burnout of teachers has been studied empirically and related to substantial interindividual differences and impaired teaching quality (relevant paper: https://doi.org/10.1016/j.chb.2020.106677). This research should be considered in the theoretical background, in particular already in the lines 88 and following (but also in section 2.1; this is currently rather underdeveloped). The research aims proffered by the authors extend this research well.
  5. Table 1 should avoid abbreviations in column 2 if possibly (or at the very least explain them).
  6. I wondered why the analyses weren’t conducted on the latent level to better account for measurement error.
  7. Exact p values and confidence intervals of the coefficients should be reported.
  8. The statistical tests for comparing coefficients between the low and high burnout subsample need to be reported.

Author Response

Ms. Ref. No.: ijerph-1504738

Title: EXPLORING HIGH VS LOW BURNOUT AMONGST PUBLIC SECTOR EDUCATORS:

COVID-19 ANTECEDENTS AND PROFILES

Response to Reviewer #3

First, let us thank you for your constructive comments. Below, each of your comments to the previous version is repeated in conjunction with our responses. Your comments are bold faced.

I have read this manuscript on burnout in public sector educators with interest. Overall, it is well written, fits well to the special issue, and makes a meaningful contribution to the literature.

Thank you for your encouragement!

There are, however, a few issues that should be considered before being able to accept this paper for publication.

  1. Abstract: Should contain information about the sample size and the analyses.

Thank you very much for your comment. The new version of the manuscript includes information about the sample size and the analyses.

  1. Generally, I wondered whether the distinction burnout VS “those that were able to develop resilience” fits well to (a) what is theoretically feasible (not ending up in burnout doesn’t necessary mean having developed resilience) and (b) what was empirically investigated (high vs low burnout levels).

Again, thank you very much for your comment. We have developed analyses for 2 different samples: high versus low burnout levels. Based on previous literature, we believe that low burnout people could be considered as resilient people. We have now explained it in the paper as follows: “The sample was divided into two groups based on their mean score on the burnout scale. We used a split that is further explained in the results section (hereafter), but we wish to stress the importance of configurating the low burnout people (those who have low score on burnout), that we propose to label them as “resilient people”. Resilience is an emerging concept and for many, represent the other pole of stress. Resilience in-volves two defining elements—the experience of adversity and positive adaptation despite this adversity [84]. First, entities must experience adversity to demonstrate and build resilience. Adversity ranges on many continuums, including severity, duration, and predictability, which impacts individuals’ ability to demonstrate resilience. What matters, is how people perceive the adversity. Resilient people view adversity as a springboard for growth and development as opposed to a negative experience that should be avoided at all costs.” We can only hope that the reviewer agree on our proposal.

  1. The introduction is a bit lengthy. I would find it helpful to already situate the present study and the research aims in the very first paragraph. Doing so, would help the readers to make more sense of the following.

Again, thank you very much for your comment. We have now presented the general aim of the paper in its first paragraph.

  1. Multiple studies have already been conducted on stress experiences and burnout during COVID19. In particular, burnout of teachers has been studied empirically and related to substantial interindividual differences and impaired teaching quality (relevant paper: https://doi.org/10.1016/j.chb.2020.106677). This research should be considered in the theoretical background, in particular already in the lines 88 and following (but also in section 2.1; this is currently rather underdeveloped). The research aims proffered by the authors extend this research well.

Again, thank you very much for your comment. We have now included this paper in our research and cite it from the first paragraph of the introduction.

  1. Table 1 should avoid abbreviations in column 2 if possibly (or at the very least explain them).

Again, thank you very much for your comment. Since we don’t have space to include the full names of the variables in the table (we’d have to decrease the size of numbers too much), we have included the explanation as a footnote.

  1. I wondered why the analyses weren’t conducted on the latent level to better account for measurement error.

Again, thank you very much for your comment. We have proceeded as in many other papers published in A* journals.

  1. Exact p values and confidence intervals of the coefficients should be reported.

Again, thank you very much for your comment. As in previous studies, we have not reported them due to space limitations. However, we could present them as supplement material.

  1. The statistical tests for comparing coefficients between the low and high burnout subsample need to be reported.

Again, thank you very much for your comment. We have now included statistical tests for comparing coefficients between the low an high burnout subsamples.

We very much appreciate your comments. In following your recommendations, we were able to substantially improve the quality of the manuscript. Thanks.

Reviewer 4 Report

This article entitled "Exploring High vs. Low Burnout Amongst Public Sector Educa-2 tors: Covid-19 Antecedents And Profiles" does provide a unique understanding regarding the work stress in the COVID era and uniquely approached the results. I do congratulate the authors for their efforts spent in studying this important issue. 

Further Comments:

1. The main concern and the question of the study is: What are the main reasons for high and low burnout among the teachers employed in the public sector during the COVID era?

2. Do you consider the topic original or relevant in the field, and if so, why? I think the work is original and the consecutive and comprehensive literature review is the evidence, which you can identify how the research tried to locate itself within.

3. What does it add to the subject area compared with other published material? The research is one of the studies in the field of stresses concerning the employees, while this study targeted the teachers in the public sector in a specific region. The methodology, the population and the and the way the study approached the problem made it contribute to the field.

4. What specific improvements could the authors consider regarding the methodology? I think the methodology is appropriate for this study.

5. Are the conclusions consistent with the evidence and arguments presented and do they address the main question posed? The conclusion is summarized what has been done including referring to and answering their main question.

6. Are the references appropriate? I guess so, the only thing I noticed one of the authors (Dolan) cited his/her too much (12 studies).

7. Please include any additional comments on the tables and figures. Tables and figures are well-located and related to the flow of the argument.

Author Response

Ms. Ref. No.: ijerph-1504738

Title: EXPLORING HIGH VS LOW BURNOUT AMONGST PUBLIC SECTOR EDUCATORS:

COVID-19 ANTECEDENTS AND PROFILES

Response to Reviewer #4

First, let us thank you for your constructive comments. Below, each of your comments to the previous version is repeated in conjunction with our responses. Your comments are bold faced.

This article entitled "Exploring High vs. Low Burnout Amongst Public Sector Educa-2 tors: Covid-19 Antecedents And Profiles" does provide a unique understanding regarding the work stress in the COVID era and uniquely approached the results. I do congratulate the authors for their efforts spent in studying this important issue. 

Thank you very much for your encouragement!

Further Comments:

  1. The main concern and the question of the study is: What are the main reasons for high and low burnout among the teachers employed in the public sector during the COVID era?

Thanks for the summary. Yes, that is our main question.

  1. Do you consider the topic original or relevant in the field, and if so, why? I think the work is original and the consecutive and comprehensive literature review is the evidence, which you can identify how the research tried to locate itself within.

Thank you!

  1. What does it add to the subject area compared with other published material? The research is one of the studies in the field of stresses concerning the employees, while this study targeted the teachers in the public sector in a specific region. The methodology, the population and the and the way the study approached the problem made it contribute to the field.

Thank you!

  1. What specific improvements could the authors consider regarding the methodology? I think the methodology is appropriate for this study.

Thank you!

  1. Are the conclusions consistent with the evidence and arguments presented and do they address the main question posed? The conclusion is summarized what has been done including referring to and answering their main question.

Thank you!

  1. Are the references appropriate? I guess so, the only thing I noticed one of the authors (Dolan) cited his/her too much (12 studies).

Thank you! We have revised the paper trying to eliminate some of Dolan citations. However, this is one of the main authors in this topic and his contributions are really important for the paper.

  1. Please include any additional comments on the tables and figures. Tables and figures are well-located and related to the flow of the argument.

Thank you!

We very much appreciate your comments. In following your recommendations, we were able to substantially improve the quality of the manuscript. Thanks.

Round 2

Reviewer 2 Report

The article doesn't have separate Methods and Results sections, with Statistics in Methods section.
The conclusion of the article does not have any with here presented results, and is simple repetition of some lines in Discussion.

An unvalidated scale for burnout measurement was used. There is repetition of data in 3 very similar and inter-correlated scales, while only one of them is used in the latest revision on Maslach Burnout Inventory.

The data should be better presented, and probably some data should go to supplementary material.

Introduction/Brief literature review should be shorten (and should be only one section - Introduction).

Author Response

Ms. Ref. No.: ijerph-1504738

Title: EXPLORING HIGH VS LOW BURNOUT AMONGST PUBLIC SECTOR EDUCATORS:

COVID-19 ANTECEDENTS AND PROFILES

Response to Reviewer #2

First, let us thank you for another round of constructive comments. Below, each of them is answered and has been synthesized with previous version. Your comments are bold faced.

The article doesn't have separate Methods and Results sections, with Statistics in Methods section.

Thank you for your comment. We have now separated methods from results.

The conclusion of the article does not have any with here presented results, and is simple repetition of some lines in Discussion.

Thanks again. We have rewritten the conclusions section. We can only hope that you like the new version.

An unvalidated scale for burnout measurement was used.

Thanks again. Based on you concern, we have included this point in the limitation of the paper. Even more, we have included more explanation about the advantages of our scale. We have explained that the scale was content validated in the past 10 years (working and applying it to executives in different languages and culture). It’s true that in this article it is also validated empirically. In addition, the authors have validated it with MBA students. Compared this scale with the MBI, the correlations is 0.83 and over .93 with the EE dimension of the MBI.

There is repetition of data in 3 very similar and inter-correlated scales, while only one of them is used in the latest revision on Maslach Burnout Inventory. The data should be better presented, and probably some data should go to supplementary material.

Thank you for your comment. As we explained you in the previous round, one of the contributions of our paper is to present the 3 dependent variables and explain that the results are consistent. For this reason, we have kept the three regression analyses. However, based on yours and reviewer 3 comments, we have now presented better the data.

Introduction/Brief literature review should be shorten (and should be only one section - Introduction).

Thank you for your comment. Based on your comments, we have eliminated the brief literature review. We have reviewed the rest of the manuscript and agree with reviewer 2 in that it was a bit repetitive with the introduction and methods sections.

We very much appreciate your comments. In following your recommendations, we were able to substantially improve the quality of the manuscript. Nonetheless, please be advised that the earlier version was copyedited by a professional copyeditor. However, we have reviewed the language again. Thanks.

Reviewer 3 Report

The authors have revised the manuscript according to the issues drawn out by me and the other reviewers. Overall, the quality of the manuscript has improved. However, there are a few issues that still warrant attention. As described in my last review, it would be more adequate to conduct the analyses on a latent level (especially taking the internal consistencies reported by the authors into consideration). If this is not done, a compelling reason needs to be included in the manuscript for justification. Further, reporting confidence interval and p values is not a matter of space limitations, as those are reported in the tables, but a matter of following proper scientific standards.

Author Response

Ms. Ref. No.: ijerph-1504738

Title: EXPLORING HIGH VS LOW BURNOUT AMONGST PUBLIC SECTOR EDUCATORS:

COVID-19 ANTECEDENTS AND PROFILES

Response to Reviewer #3

First, let us thank you for another round of constructive comments. Below, each of them is answered and has been synthesized with previous version. Your comments are bold faced.

The authors have revised the manuscript according to the issues drawn out by me and the other reviewers. Overall, the quality of the manuscript has improved. However, there are a few issues that still warrant attention.

Thank you very much for your comment. We really appreciate that the reviewer has seen an improvement.

As described in my last review, it would be more adequate to conduct the analyses on a latent level (especially taking the internal consistencies reported by the authors into consideration). If this is not done, a compelling reason needs to be included in the manuscript for justification. Further, reporting p values is not a matter of space limitations, as those are reported in the tables, but a matter of following proper scientific standards.

Again, thank you very much for your comment. Our analyses are conducted on a latent level and we have now reported p values in tables 2, 3 and 4.

We very much appreciate your comments. In following your recommendations, we were able to substantially improve the quality of the manuscript. Thanks.
